# MULTI-MODEL INDUCED SOURCE-FREE VIDEO DOMAIN ADAPTATION

## ABSTRACT

Existing Source-free Video Domain Adaptation (SFVDA) aims to learn a target video model for an unlabeled target domain by transferring knowledge from a labeled source domain using a single pre-trained source video model. In this paper, we explore a new SFVDA setting where multiple source domains exist, each offering a library of source models with different architectures. This setting offers both opportunities and challenges: while the presence of multiple source models enriches the pool of transferable knowledge, it also increases the risk of negative transfer due to inappropriate source knowledge. To tackle these challenges, we introduce the **Multiple-Source-Video-Model Aggregation Framework (MSVMA)**, comprising two key modules. The first module, termed **Multi-level Instance Transferability Calibration (MITC)**, enhances existing uncertainty-based transferability estimation metrics by incorporating scale information from both group and dataset levels. This integration facilitates accurate transferability estimation at the instance level across diverse models. The second module, termed **Instance-level Multi Video Model Aggregation (IMVMA)**, leverages the calculated instance-level transferability to guide a path generation network. This network produces instance-specific weights for unsupervised aggregation of source models. Empirical results from three video domain adaptation datasets demonstrate the state-of-the-art performance of our MSVMA framework.

## 1 INTRODUCTION

Video action recognition is a crucial task in video understanding, which has continually garnered attention and research due to its general applicability. Recently, significant advancements in video action recognition have been achieved with deep learning(1; 2; 3), largely due to the emergence and availability of large-scale labeled datasets(4; 5). However, the construction of large-scale labeled

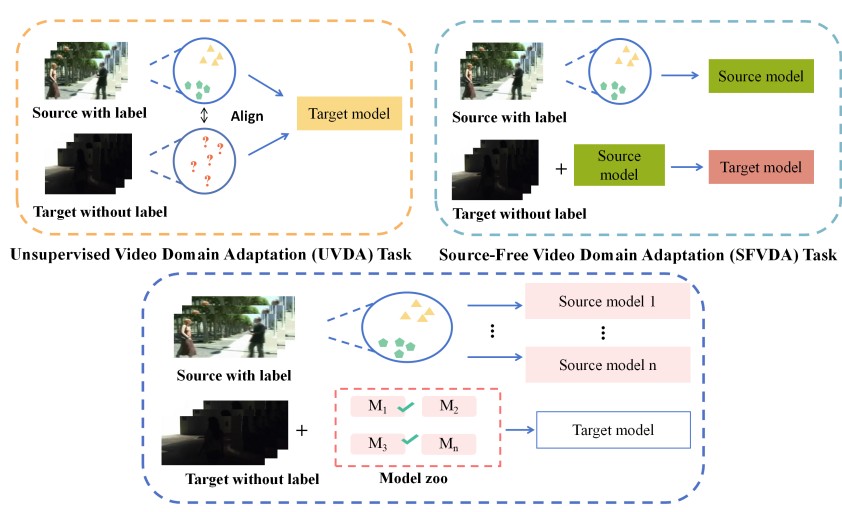

Figure 1: Illustration of the UVDA , SFVDA and MSFVDA task settings.

datasets for real-world scenarios incurs substantial manpower and financial costs, which poses a significant challenge when adapting models to various sceneries.

Unsupervised Video Domain Adaptation (UVDA) has been proposed recently (6), which aims to transfer knowledge from a labeled video dataset (source domain) to an unlabeled target domain. Existing approaches address this task by minimizing the domain discrepancy between the source and target domain based on adversarial training (7; 8; 9; 10) and self-supervised learning methods (11; 12; 13).

While traditional UVDA methods effectively mitigate domain shift between different video sources, they necessitate access to source data during the adaptation step, which poses significant risks of privacy breaches and incurs substantial data transmission costs. To address these challenges, Source-Free Video Domain Adaptation (SFVDA) has emerged as a promising alternative (14). SFVDA relies solely on a single pre-trained model from a labeled source domain to learn an action recognition model for the unlabeled target domain, without accessing the original source videos. Recently, a state-of-the-art SFVDA method based on temporal and spatial consistency has been proposed (15). This method adapts the source model to learn the capabilities of motion dynamics and action coherence in videos by applying temporal and spatial augmentations to simulate domain transitions.

A primary limitation of current Source-Free Video Domain Adaptation (SFVDA) methods is their reliance on a single source domain model. However, numerous source models from different sources with diverse architectures are generally available in real-world scenarios, which offer a wide variety of knowledge. This has motivated us to explore more comprehensive adaptation strategies by integrating information from multiple source domain models, a method we termed Multi-model Source-Free Video Unsupervised Domain Adaptation (MSFVDA). By providing a zoo of well-trained source models with various architectures from the source domain, target users can access and leverage multiple models for domain adaptation, thereby enhancing the knowledge base of the source domain. However, a critical challenge in developing MSFVDA lies in the effective aggregation of the multiple source domain models.

To best of our knowledge, this problem has not yet been explored in the video domain. Beyond the video task, a closely related work is SUTE (16), which address multi-model adaptation in image task. The key of this method is to estimate the transferability of each model and select models for aggregation based on the estimated transferability. This method operates at the dataset-level, i.e., estimating the transferability of a model on the entire dataset. However, due to the inherent spatial and temporal complexities in videos, significant variability exists among video instances. Consequently, estimating model transferability solely at the dataset-level fails to account for the variability of models toward individual instances. Recently, there is a study that focuses on the multi-model aggregation at instance-level (17). However, this method relies on labeled data for path weight learning, which is unsuitable in MSFVDA.

In this paper, we introduce a Multiple-Source-Video-Model Aggregation (MSVMA) framework to address the MSFVDA. To resolve the challenge of instance-level transferability estimation, we propose a novel Multi-level Instance Transferability Calibration (MITC) algorithm. MITC seeks to measure the instance-level transferability based on the uncertainty methods (18). Therefore, we introduce a novel calibrate function, which further calibrates the uncertainty-based instance-level transferability by incorporating scale information from both group-level and dataset-level. For more effective instance-level aggregation, we introduce an Instance-level Video Multi-Model Aggregation (IMVMA), which accounts for the significant differences among video instances. IMVMA learns to assign instance-level weights for each video instance in the path generation network based on the instance-level transferability estimated by MITC, and then selectively activate source domain models to achieve instance-level model aggregation. We demonstrated the effectiveness of our method on three public datasets and achieved state-of-the-art results.

Our contributions can be summarized as follows:

1. We propose a novel multi-level instance transferability calibration algorithm (MITC), which leverages transferability information across multiple scales to calibrate instance-level transferability. This approach enables more accurate instance-level source-free video transferability measurement;

2. We develop a new Instance-level Multi Video Model Aggregation (IMVMA) framework assisted by our proposed MITC. By integrating multiple source models from source domains

with varied architectures, IMVMA gathers more comprehensive knowledge and achieves better accuracy and stability for domain adaptation;

3. We test our model on the Daily-DA, Sports-DA, and UCF-HMDB$_{full}$ datasets for video action recognition. All the results support that our MSFVDA model brings a large performance boosting compared to other state-of-the-art models.

## 2 RELATED WORK

**Source Free Unsupervised Video Domain Adaptation.** Image-based Source-Free Unsupervised Domain Adaptation (SFDA) has garnered significant attention recently (19; 20; 21). The primary goal of SFDA is to adapt models trained on a source domain directly to an unlabeled target domain without requiring access to source domain data. In contrast, Source-Free Video Domain Adaptation (SFVDA) has only recently begun to be explored. The unique temporal properties inherent to video data present a significant challenge for SFVDA. Xu et al. (22) proposed a SFVDA method that leverages the temporal properties of video data based on temporal consistency. Similarly, Li et al. (15) also took advantage of temporal consistency, by exploring the model's self-adaptive capabilities in both temporal and spatial information. However, previous studies have primarily focused on the adaptation issues of single models, heavily restricting the overall performance. Recently, Pei et al. (16) introduce a transferability measure to assist in model selection, assigning dataset-level weights during aggregation, and achieving excellent performance in various image recognition tasks. Building on this, our work proposes a new instance-level video multi-model aggregation framework that can simultaneously learn accurate path weights in an unsupervised manner and assist in model selection. This approach further enhances the effectiveness of multi-model aggregation.

**Transferability Measurements.** In scenarios where multiple pre-trained source domain models are available, it is particularly crucial to evaluate the transferability of each source domain model to the target domain. A traditional way predicts the performance of the source model after fine-tuning in a supervised manner on the target domain (23; 24; 25; 26). However, the requirement for target labels in these methods above hinders their application in broader contexts. Recently, some methods are proposed for estimating the transferability for source-free unsupervised tasks (27; 28). These method can be roughly divided into two classes: distribution-induced methods and uncertainty-based methods. Distribution-induced methods (16; 29) measure transferability by developing and evaluating some distribution-related assumption. For example, SUTE (16) proposed hypotheses on dataset distribution, including Individual Certainty, Semantic Consistency, and Global Dispersion. MDE (29) proposes to levegate the energy hypothesis and converts the information of all samples into a statistical probability distribution. However, estimating transferability solely from the perspective of dataset distribution overlooks the model's variability in individual instances. Uncertainty-based methods leverage uncertainty methods for transferability estimation, including entropy, temporal and spatial disturbances, based on the assumption that high transferable model exhibits low uncertainty (30; 18). However, uncertainty-based methods fail to accurately estimate transferability across different models (19). To address this, we propose the MITC method to perform instance-level and cross-model transferability estimation.

## 3 METHOD

### 3.1 PROBLEM DEFINITION AND NOTIONS

Assume the source domain $D_S$ contains $|D_S|$ labeled videos $\{(V_i, y_i)\}_{i=1}^{|D_S|}$ and provides $M$ adaptive video classification models $h_i$. The target domain $D_T$ contains $|D_T|$ unlabeled videos denoted as $\{V_i\}_{i=1}^{|D_T|}$. The final goal of MSFVDA is to learn a target model $H = \{h_i|, 1 \leq i \leq M\}$ from the $M$ source models and apply it to the target domain video. In this paper, we represent the feature extraction for each target video by $f(V_i)$, and the output of each target video as $h(V_i)$.

### 3.2 SOURCE MODEL GENERATION

Unlike the existing SFVDA methods, we conduct the training of the source model directly based on the mmaction2 (31) framework. We separate the model $h$ into two components: $h = F \circ C$,

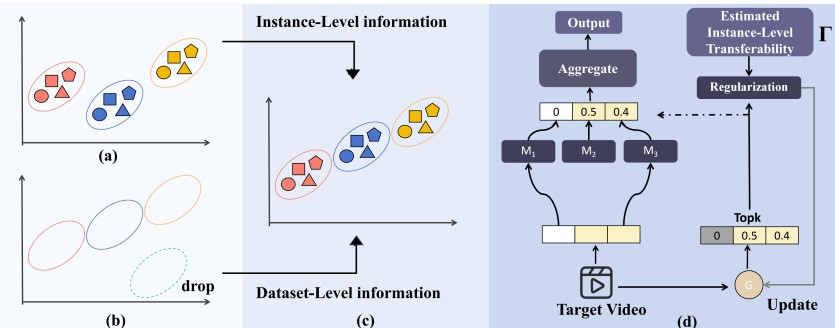

Figure 2: Illustration of our Multiple-Source-Video-Model Aggregation (MSVMA) framework: (a) Current uncertainty-based methods focus on model transferability but lack generalization across models. (b) Distribution-induced methods estimate dataset-level transferability across models but neglect instance-level transferability within models. (c) To overcome these limitations, we propose a Multi-level Instance Transferability Calibration (MITC) approach that accurately calibrates instance-level transferability using multi-dimensional information. (d) Our Instance-Level Multi-Video Model Aggregation (IMVMA) framework employs a path generation network to assign customized instance-level weights, with MITC ($\Gamma$) adjusting these weights unsupervised to ensure accurate distribution. The aggregated model is then used as the target model.

where $F$ serves as the feature extractor encapsulating the temporal data within the feature, and $C$ functions as the classifier. For each video input $V_i$, it passes the feature extractor to get a vector representation for $V_i$, which will be sent to $C$ for calculating the cross-entropy loss with $y_i$. $h$ is updated accordingly with the standard gradient back-propagation.

### 3.3 ANALYSIS OF EXISTING TRANSFERABILITY MEASUREMENTS

While MSFVDA provides a more extensive knowledge base from the source domains, incorporating multiple distinct source models may increase the risk of including those that underperform in the target domain, thereby causing a substantial decline in overall performance. The experimental results demonstrate that careful model selection, rather than indiscriminately using all available source models, significantly improves adaptation performance in the target domain. These findings highlight the critical need for effective transferability estimation metrics within the MSFVDA framework to accurately assess and select the most suitable source domain models.

Existing transferability estimation methods can be roughly categorized into two groups: distribution-induced methods (16; 29) that develop and evaluate some distribution-related assumption and uncertainty-based methods which are based on the assumption that high transferable model exhibits low uncertainty (30; 18).

While these methods demonstrate effectiveness in image model transferability measurement tasks, we observed the experiment result in table 1 that both methods are not suitable in our MSFVDA, which are discussed in the following.

*Limitation of distribution-induced methods:* These methods utilize dataset-level information for cross-model transferability estimation but overlook the fact that source domain models may exhibit different preferences for individual instances. Unlike images, videos consist of a series of consecutive frames that contain both spatial and temporal information. The temporal dimension introduces additional complexity and variability. Moreover, videos typically have greater content complexity compared to images, resulting in more significant differences between video instances than between image instances. Ignoring instance-level transferability estimation can hinder further improvements in adaptation performance.

*Limitation of Uncertainty-based methods:* These methods can estimate instance-level transferability within a single model. However, when the repository includes models with diverse architectures, differences between these models introduce bias at the dataset-level. Without addressing these cross-model differences, directly estimating instance-level transferability across different source domain models may not provide accurate results.

## 3.4 INSTANCE-LEVEL TRANSFERABILITY CALIBRATION

The above discussion highlights two key findings:

- **Existing distribution-induced methods accurately estimate transferability at the dataset-level (coarse-grained) across different models**, but they do not extend to fine-grained, instance-level transferability assessments.
- **Current uncertainty-based methods provide reliable metrics for instance-level (fine-grained) transferability estimation within the same model.** However, they lose efficiency when estimating transferability across different models.

Based on these findings, we propose a novel calibration function that integrates the strengths of both distribution-induced and uncertainty-based methods, thus enabling accurate instance-level transferability estimation across different models. Specifically, the calibration function is formulated by:

$$\Phi(a, b) = \frac{a}{a_{\text{norm}}} \left(1 + \ln\left(1 + b\right)\right) \tag{1}$$

where $a$ represents fine-grained transferability, while $b$ represents large-scale information. This calibration function is governed by the following two principles:

- During calibration, it is crucial to consider both fine-grained transferability and larger-scale information to properly align the transferability estimates across source domain models with different architectures.To preserve the relative relationships between instances within a model, we focus on fine-grained transferability as the primary measure. By normalizing this measure, we eliminate the scale discrepancies caused by different model architectures, allowing for comparable instance-level transferability across models. This approach helps identify the model preferences for the current instance.

- Scale information is provided by coarse-grained transferability. Coarse-grained transferability is effective for cross-architecture model evaluation. However, because it does not account for differences between instances at the instance-level, it should only serve as a scale reference for comparisons between models.

**Instance Transferability Calibration.** In this paper, we employ the previously developed Source-Free Transferability Estimation (SUTE) (16) as our coarse-grained transferability measurements (terms $\mathcal{T}_\mathcal{D}$ ) due to its efficiency. The calculation method for SUTE is as follows:

$$\text{SUTE} = \mathbb{E}_{\mathbf{V_i} \sim \mathbf{D}} \mathcal{H}(h(V_i)) - \mathcal{H}(P_{\tilde{y}|\hat{y}}) + \mathcal{H}(\mathbb{E}_{\mathbf{V_i} \sim \mathbf{D_T}}(h(V_i))) \tag{2}$$

where $\hat{y}$ denotes predictive semantics, $\tilde{y}$ denotes the pseudo label for each target data,and $\mathbb{E}$ denotes the expected value. We additionally add a piecewise function (parameterized by $\tau$) of SUTE to formulated our $\mathcal{T}_\mathcal{D}$, formulated by:

$$\mathcal{T}_\mathcal{D} = \gamma(\text{SUTE}; \tau) = \begin{cases} \text{SUTE}, & SUTE \geq \tau. \\ -\infty, & \text{SUTE} < \tau. \end{cases} \tag{3}$$

This is because we observed that models with very small SUTE values exhibit poor transferability. Aggregating such models with others significantly affects the adaptation results.

Then, we adopt the entropy as our fine-grained transferability measurement. Given an input instance $V_i$, it is formulated by $\mathcal{T}_\mathcal{I} = \mathcal{H}(h(V_i))$. Finally, by utilizing the proposed calibration function (Equation 1), the calibrated instance-level transferability is formulated by $\Phi(\mathcal{T}_\mathcal{I}, \mathcal{T}_\mathcal{D})$.

**Multi-Level Instance Transferability Calibration.** The core idea of Instance Transferability Estimation Calibration is to calibrate fine-grained (i.e., instance-level) transferability based on the scales provided by coarse-grained (i.e., dataset-level) transferability, since the latter is easier. Building on this idea, we further introduce an intermediate grouping, referred to as the "group-level," to enhance the transferability calibration. A "group" is defined as a set of the $k$ Nearest Neighborhoods of a sample within the feature space. Hence, this level serves as a finer granularity compared to the dataset-level, while remaining coarser than the instance-level. In this paper, we formulate the group-level transferability as the maximum distance between samples within group. This implicitly

reflects the characteristic that, for a transferable model, samples belonging to the same class should be closer in the feature space. Specifically, the group-level transferability is formulated by:

$$\mathcal{T}_{\mathcal{G}} = \max \left\{ d(f(V_i), f(V_{j^*})) \,\Big|\, j^* \in \arg\operatorname{sort}_k \big( d(f(V_i), f(V_{j^*})) \big) \right\} \tag{4}$$

The $\arg\operatorname{sort}_k$ function returns the indices that would sort the distances in ascending order and selects the top $k$ indices.

Thus, our Multi-Level Instance Transferability Calibration (MITC) framework first calibrates group-level transferability (denoted as $\mathcal{T}_{\mathcal{G}}$) using dataset-level transferability, represented as $\Phi(\mathcal{T}_{\mathcal{G}}, \mathcal{T}_{\mathcal{D}})$. Next, we calibrate instance-level transferability based on the calibrated group-level transferability. The complete MITC formulation is as follows:

$$\text{MITC} = \Phi(\mathcal{T}_{\mathcal{I}}, \Phi(\mathcal{T}_{\mathcal{G}}, \mathcal{T}_{\mathcal{D}})) \tag{5}$$

### 3.5 Instance-level Source Model Aggregation

Aggregation of multiple source domain models can be facilitated by the MSFDA methods (32; 15; 33; 34), which derive domain-level integration weights and apply these uniformly across all target instances. Although the learned weights offer an intuitive interpretation based on domain transferability, they inevitably introduce misalignment and bias at the instance level. Moreover, videos, in contrast to images, exhibit more pronounced misalignment and bias at the instance level, as illustrated in Fig. 2. Directly assigning a fixed weight to the model undeniably affects performance. This prompts us to explore dynamically assigning aggregation model weights to different instances during model aggregation. Inspired by the prior concept of pathways in deep networks, different input videos have distinct preferences for different source domain models, activating various source domain models and assigning them different weights. To this end, we implement a pathway generation network $G$, which outputs data-dependent pathway weights $G(V_i)$, where each dimension represents the weight of a specific model in the hub. To utilize the most suitable pre-trained models for the target data, we retain only the top k pathway weights and set the remaining pathway weights to zero:

$$G(V_i) = f_{\text{topk}}(G(V_i), k) \tag{6}$$

where $f_{\text{topk}}(G(V_i), k)_j$ is defined as $G(V_i)_j$, if $G(V_i)_j$ is among the top $k$ values of $G(V_i)$; otherwise, it is defined as 0.

Based on the generated path weights $G(V_i)$, we only pass the input data to the pre-trained models where the path weights are greater than zero. At the same time, considering that the path generation network operates under an unsupervised context for learning and generating path weights, there is an inherent risk in directly using the weights produced by the path generation network. To address this issue, we enhance the learning resistance to smooth the path weights. Furthermore, in order to let the pathway generation network generate an accurate route in an unsupervised setting, we utilize the multi-level instance transferability calibration (MITC) to assist the pathway generation network in learning more reasonable instance-level weights. This is achieved by calculating the L2 loss between the instance-level weights provided by $G(V_i)$ and MITC where $\text{MITC} = (\text{MITC}_1, \ldots, \text{MITC}_{|\mathcal{S}_i|})$.

$$L_{cor} = \frac{1}{n} \sum_{i=1}^{n} (\Gamma_i - G(V_i)) \tag{7}$$

$\Gamma_i$ reperesents the MITC of the $i$-th source model, to constrain the path weights $G(V_i)$. Then the final output of the video instance-level Source Model aggregation framework is:

$$output = A \left( [G(V_i) \cdot h_i(V_i)]_{i=1}^{k} \right) \tag{8}$$

where A composes the outputs from different source selected models and function $[\cdot]$ concatenates all the path selected model outputs vectors.

### 3.6 Overall Learning Objective

In the adaptation process, we employ the SHTC method and simultaneously constrain the weights generated by the path generation network during training using the $\mathcal{L}_{\text{cor}}$ and $\mathcal{L}_{SHTC}$ (15) adopts

from the SHTC method. Our final learning objective is:

$$\mathcal{L} = \mathcal{L}_{SHTC} + \theta_1 \mathcal{L}_{cor} \tag{9}$$

where $\theta_1$ are tradeoff hyperparameters.

## 4 EXPERIMENTS

### 4.1 EXPERIMENTAL SETTINGS

**Datasets.** We conducted experiments on three common benchmark datasets. Benchmark including: 1) UCF-HMDB$_{full}$ comprises videos from 12 overlapping classes from the UCF101 (U) dataset (35) and the HMDB51 (H) dataset (36). 2) UCF-Sports-1M is derived from the SportsDA benchmark, which included two datasets: UCF101 (U) (35) and Sports-1M (S)(37). 3) Daily-DA is another large-scale cross-domain action recognition benchmark. We excluded Kinetics from our experiments AS the pre-trained models we used from the mmaction2(31) framework were pre-trained on Kinetics-400. Resulting in three datasets: ARID (A) (38), HMDB51 (H) (36), and Moments-in-Time (M) (5). The detailed information are listed in the supplementary materials. Among them, the Daily-DA and UCF-Sports1M datasets, when used for adaptation, utilize the data according to the domain-specific test sets partitioned as described in the TAMAN(39).

**Implementation details.** Details of the specific implementation can be found **in the supplementary materials.**

**Baseline methods.** Our primary comparisons are with SHOT (32), STHC (15), DECISION (33), CAiDA (34), and KD3A (33). Among these, SHOT is a classical method in source-free domain adaptation, STHC is the state-of-the-art method in source-free video domain adaptation. DECISION, CAiDA, and KD3A are designed as state-of-the-art methods for multi-source free domain adaptation(MSFDA). For instance-level transferability estimation, we primarily compared our approach with several methods adapted from Distribution-induced methods. These include Negative Mutual Information(NMI) (40; 32), pseudo-label-based methods LEEP (26)/LogME (23) (referred to as LEEP∗/LogME∗ at the instance-level), the energy-based method Meta-Distribution Energy (MDE) (29), and the source-free unsupervised transferability estimation metric (SUTE) (16). Additionally, we compared our method with directly using uncertainty-based approaches, including entropy, temporal consistency, and spatial consistency(30; 18).

### 4.2 PERFORMANCE COMPARISON AND ANALYSIS

**Comparative Studies.** Table 1 shows the correlation between instance-level transferability estimation methods and instance prediction accuracy. We extend existing dataset-level transferability metrics (distribution-based methods) to the instance level, but found that their performance are suboptimal. For instance, methods like MDE failed in certain tasks, with the highest average correlation of other methods being only 0.184. Additionally, current instance-level methods (uncertainty-based methods) also performed poorly due to structural differences among models; for example, consistency methods failed, and the entropy method achieved an average correlation of only 0.204.

Our proposed method, MITC, addresses these issues by leveraging multi-scale information to refine instance-level transferability within models, enabling more accurate cross-model comparisons. This approach resulted in significant improvements across three public datasets: **0.268 improvement on Daily-DA, 0.022 on Sport-DA, and 0.033 on UCF-HMDB$_{full}$**. The only exception was the S→U task in the Sports-DA dataset, where the minimal domain shift led to similar transferability across source domain models, allowing uncalibrated instance-level transferability to accurately reflect the true transferability of instances.

Table 2 presents test results on the more challenging Daily-DA dataset. Using multiple source domain models provides richer information, leading to better adaptation than a single model. However, not all source domain models are highly transferable. Directly aggregating all models can degrade performance. For example, for SHOT, when using SUTE for model selection before aggregation, it outperforms direct aggregation of all source models by an average of 2.81%. We found that selecting models with high transferability based on dataset-level estimation improves the aggregated model's performance compared to using all source domain models.

Table 1: Spearman rank correlation coefficient on Daily-DA, Sports-DA and UCF-HMDB$_{full}$. Spearman rank correlation coefficient between the cross entropy loss of video instance on the target domain and the measured instance transferability under the **MSFVDA** setting. IL denotes whether the method estimate transferability on instance level. "-" denotes that the results do not have statistics significance(re., $p$-value $> 0.05$). Best results are in bold font.

| Method | IL | Daily-DA | | | | | | | Sports-DA | | | UCF-HMDB$_{full}$ | | |
|---|---|---|---|---|---|---|---|---|---|---|---|---|---|---|
| | | M→A | H→A | A→M | H→M | A→H | M→H | Avg. | U→S | S→U | Avg. | H→U | U→H | Avg. |
| NMI | × | 0.322 | 0.230 | -0.077 | 0.087 | 0.238 | 0.303 | 0.184 | 0.200 | 0.553 | 0.377 | 0.531 | 0.256 | 0.394 |
| LogME* | × | 0.200 | 0.050 | 0.249 | -0.057 | 0.212 | - | N/A | 0.157 | 0.323 | 0.240 | 0.154 | 0.111 | 0.133 |
| LEEP* | × | 0.341 | 0.211 | -0.327 | - | 0.305 | 0.185 | N/A | 0.179 | 0.100 | 0.140 | 0.122 | - | N/A |
| MDE | × | 0.183 | 0.188 | -0.211 | 0.068 | -0.232 | - | N/A | -0.214 | 0.249 | 0.018 | -0.082 | 0.247 | 0.083 |
| SUTE | × | 0.354 | 0.228 | 0.310 | 0.127 | 0.382 | 0.290 | 0.282 | 0.198 | 0.268 | 0.233 | 0.259 | 0.175 | 0.217 |
| Entropy | √ | 0.388 | 0.302 | -0.202 | 0.215 | -0.040 | 0.558 | 0.204 | 0.687 | **0.982** | 0.835 | 0.932 | 0.740 | 0.836 |
| Temporal consistency | √ | 0.139 | -0.090 | 0.066 | -0.044 | 0.081 | - | N/A | 0.121 | -0.034 | 0.044 | 0.332 | 0.225 | 0.274 |
| Spatial consistency | √ | 0.345 | 0.096 | 0.088 | -0.047 | 0.197 | - | N/A | - | 0.162 | N/A | 0.049 | 0.056 | 0.053 |
| MITC (Ours) | √ | **0.762** | **0.447** | **0.308** | **0.273** | **0.417** | **0.622** | **0.472** | **0.739** | 0.975 | **0.857** | **0.954** | **0.783** | **0.869** |

Next, we compared our method with SFDA, MSFDA, and SFVDA, which employ model selection techniques. We evaluated all dataset-level transferability estimation methods from Table 1 across different datasets and ultimately compared our instance-level video aggregation framework, IMVMA, with the two best-performing transferability estimation methods. Both of these methods selected the top three most transferable models for aggregation. In our framework, the path generation network only activated the two models with the highest weights. The final adaptation results on three datasets demonstrate that IVSUTE outperformed the current SFDA, MSFDA, and SFVDA methods. Additional experimental results are provided in the supplementary materials.

IMVMA focuses on significant differences between video instances, leading to a substantial performance improvement compared to methods that assign fixed weights to all models. **It achieved an average accuracy improvement of 4.29% over the second-best method and a 21.84% improvement over the average performance of individual source domain models**. Although multi-level instance transferability calibration enhances the accuracy of weight adjustment, inaccurate instance-level transferability estimation can negatively impact the path generation network's learning, thereby reducing aggregation performance. For example, in the H→M task, the method without instance-level weighting outperformed our instance-based video framework. This indicates that while MITC significantly improves instance-level transferability estimation accuracy, room for further improvement in internal model transferability to enhance MITC's effectiveness still remains.

**Tables 1 and 2 in the supplementary materials** present the test results on relatively simpler datasets, where direct adaptation of source domain models to the target domain already yielded strong performance. Aggregating all models directly resulted in significant performance gains. Nevertheless, our Multiple-Source-Video-Model Aggregation (MSVMA) framework still **achieved a 0.55% improvement over the best aggregation model on the UCF-Sports1M, and an 11.31% improvement over the average performance of individual source domain models. On the UCF-HMDB$_{full}$, the improvements were 3.69% and 17.07%**, respectively, compared to the average results of individual source domain models. Additional results are provided in the supplementary materials. Although the improvement in the S→U task using the MSVMA were not substantial, this was due to the similar performance of the source domain models in the model library, resulting in minimal differences between direct aggregation and transferability-based selection, and thus limiting the effect of instance-level weighting.

## 4.3 Ablation Studies

**Effect of Each Level Calibration.** Table 3 presents the experimental results of calibrating transferability across different hierarchical levels. When only dataset-level large-scale information is applied to calibrate intermediate-level groups, the overall performance improves by 0.09. However, performance **decreases by 0.163, 0.047, and 0.116 in the H→A, H→M, and M→H tasks, respectively**. These results suggest that fine-grained transferability estimation is crucial. Although using either group-level or dataset-level information in isolation preserves the fine-grained relationships between models and supports cross-model transferability estimation, incorporating intermediate-

Table 2: Results on Daily-DA. Source represents the average performance of the models from the source domain model zoo on the target domain. The best results are in bold.

| Method | M→A | H→A | A→M | H→M | A→H | M→H | Avg. |
|---|---|---|---|---|---|---|---|
| Source | 42.49 | 35.07 | 27.50 | 39.08 | 36.34 | 57.92 | 39.73 |
| SHOT | 58.95 | 44.56 | 43.25 | 48.50 | 62.50 | 68.33 | 54.35 |
| +NMI | 60.26 | 44.56 | 33.75 | 46.50 | 62.50 | 74.58 | 53.69 |
| +SUTE | 63.10 | 47.92 | 47.25 | 49.25 | 62.50 | 72.92 | 57.16 |
| STHC | 59.91 | 47.85 | 43.00 | 48.00 | 62.08 | 67.92 | 54.79 |
| +NMI | 60.29 | 51.31 | 44.00 | 45.75 | 60.42 | 72.08 | 55.64 |
| +SUTE | 62.87 | 51.16 | 47.75 | 49.00 | 60.00 | 72.92 | 57.28 |
| Decision | 58.35 | 46.30 | 42.75 | 49.00 | 57.92 | 67.92 | 53.71 |
| +NMI | 60.51 | 44.02 | 43.25 | **50.25** | 60.42 | 73.75 | 55.37 |
| +SUTE | 63.07 | 45.31 | 43.25 | 48.25 | 57.08 | 73.33 | 55.05 |
| Kd3A | 60.15 | 47.51 | 43.00 | 48.00 | 62.08 | 67.92 | 54.78 |
| +NMI | 60.44 | 51.25 | 43.59 | 46.00 | 60.42 | 71.67 | 55.55 |
| +SUTE | 63.50 | 51.25 | 46.00 | 49.00 | 60.42 | 72.92 | 57.18 |
| CAiDA | 58.57 | 46.06 | 42.75 | 49.00 | 61.25 | 68.33 | 54.33 |
| +NMI | 60.24 | 48.10 | 45.00 | 45.50 | 62.00 | 72.50 | 55.56 |
| +SUTE | 60.93 | 51.67 | 46.00 | 48.25 | 61.25 | 70.42 | 56.42 |
| Ours | **70.36** | **52.74** | **50.75** | 47.25 | **68.75** | **79.58** | **61.57** |

Table 3: Ablation on multi-level calibration.

| $\mathcal{I}$ | $\mathcal{G}$ | $\mathcal{D}$ | M→A | H→A | A→M | H→M | A→H | M→H | Avg. |
|---|---|---|---|---|---|---|---|---|---|
| ✓ | | | 0.388 | 0.302 | -0.202 | 0.215 | -0.040 | 0.558 | 0.204 |
| | ✓ | ✓ | 0.421 | 0.139 | 0.295 | 0.168 | 0.301 | 0.442 | 0.294 |
| ✓ | | ✓ | 0.761 | **0.447** | 0.306 | **0.273** | 0.417 | 0.621 | 0.471 |
| ✓ | ✓ | | 0.757 | 0.440 | **0.323** | 0.263 | **0.420** | 0.620 | 0.471 |
| ✓ | ✓ | ✓ | **0.762** | **0.447** | 0.308 | **0.273** | 0.417 | **0.622** | **0.472** |

level transferability information further enhances the accuracy of these estimates. This underscores the advantage of a multi-level calibration approach.

**Robustness of MITC and Effectiveness of the Calibration Function.** Figure 3a demonstrates the improvement in cross-model transferability estimation for two uncertainty-based methods, temporal consistency and spatial consistency, following the application of the MITC approach. This illustrates the robustness of MITC, indicating that its effectiveness extends beyond a single uncertainty-based transferability estimation method. Figure 3b showcases the effectiveness of the proposed calibration function. Compared to directly using dataset-level and group-level information, our calibration function proves more effective due to two key properties: it preserves the relative relationships among instances within a model, and it uses dataset-level and group-level information solely as scale references. These features significantly improve the accuracy of instance-level transferability estimation.

Table 4: Ablation study on instance-level multi-video models aggregation framework. Best results are in bold font.

| Method | M→A | A→H |
|---|---|---|
| Source only best(oracle) | 61.84 | 62.50 |
| Pathway w/o topK | 70.18 | 67.08 |
| Pathway w/o calibrate | 70.19 | 65.83 |
| Ours | **70.36** | **68.75** |

**Weight analysis.** Figure 4a demonstrates the significance of instance-level weights, highlighting that directly assigning fixed weights to the models to be aggregated is far less effective than assigning specific weights for each input instance. As shown in Table 4, we report the results of direct adaptation of the best-performing models in the source domain model zoo (Oracle), the results using the IMVMA framework without path selection, and the results using the IMVMA framework without instance-level weight calibrate for path weights.

We observed that directly learning path weights in an unsupervised setting is suboptimal. However, when all selected source domain models are activated without path selection and with cor-

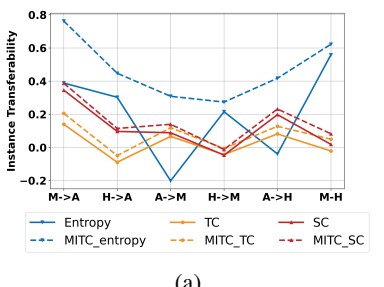 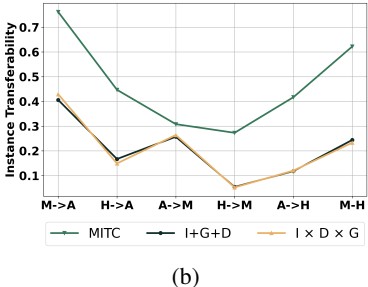

| (a) | (b) |

Figure 3: (a) Robustness of MITC. (b) Effectiveness of the calibration function.

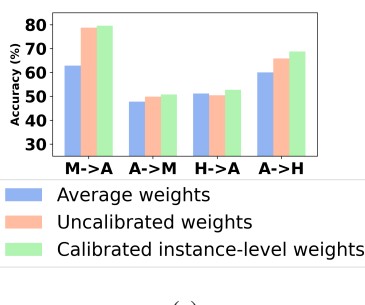 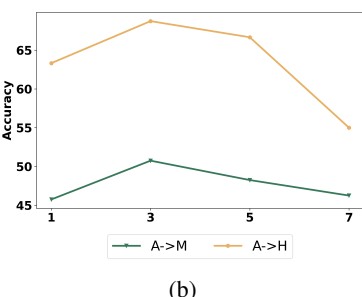

| (a) | (b) |

Figure 4: (a) Quality of instance-level weights. (b) Aggregation model number selection.

rected weights, performance surpasses that of the Oracle model. This indicates that utilizing multiple highly transferable source domain models can effectively enhance adaptation performance. Moreover, selectively activating a subset of models during aggregation and constraining unsupervised adaptation with instance-level transferability metrics can further improve performance, with **a 8.52% improvement on the M→A task and a 6.25% improvement on the A→H task compared to the Oracle model**. This suggests that in an unsupervised context, the path generation network requires the support of instance-level transferability to accurately activate the appropriate models, thereby enhancing overall dataset-level performance. The variation among video instances allows instance-level model aggregation, guided by instance-level transferability, to outperform fixed-weight model aggregation, leading to improved adaptability in the target domain.

**Number of Models for Aggregation.** When using IMVMA for model aggregation, we select the top three models with the highest transferability scores. Figure 3(b) shows that aggregating too many models can introduce poorly transferable ones, harming overall performance. Selecting only one model limits the benefits of diverse knowledge. Thus, careful selection of the number of models is essential to balance the benefits and risks of aggregation.

## 5 CONCLUSION

This paper introduces a new SFVDA setting called MSFVDA, where enables each source domain to provide a zoo of trained source models , and allows the target user to utilize any model from these model zoos without limitations on quantity. We propose a Multiple-Source Video Model Aggregation (MSVMA) framework for this setting. MSVMA employs Multi-level Instance Transferability Calibration (MITC) to integrate group-level and dataset-level scale information, improving existing uncertainty-based transferability estimation metrics. This allows for accurate instance-level transferability estimation across different models. For target domain adaptation, we introduce Instance-level Multi-Video Model Aggregation (IMVMA), which uses the calculated instance-level transferability to guide a path generation network. This network assigns instance-specific weights for unsupervised source model aggregation, achieving state-of-the-art performance on MSFVDA tasks.

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

# A  APPENDIX

Regarding the selection of source domain models, we curated a model zoo comprising 15 models for the three datasets. This includes C3d (41), SlowFast (42), various configurations of I3d (43) and SlowOnly (42), as well as different backbone of VideoSwin (44) and MViTv2 (45). The parameters for model pre-training and video clip segmentation were configured based on the settings within the mmaction2 framework. All experiments were conducted on an NVIDIA A100. The pathway generation network was implemented using C3d. The hyperparameter $\theta_1$ in equation 8 was set to 0.01. Instance-level transferability was evaluated by calculating Spearman's rank correlation coefficient (46) between the cross-entropy of each source model on target domain instances and the estimated transferability for each instance. We report the Mean-1 accuracy on the target domain, which represents the average class accuracy, averaged over five runs for each method under the same setup.

THe various configurations of I3d includes: The I3D model includes different backbone networks: ResNet50 (NonLocalDotProduct), ResNet50 (NonLocalEmbedGauss), and ResNet50 (NonLocal-Gauss); along with two different sampling strategies: 32x2x1 and dense-32x2x1. The Slowonly model includes different backbone networks: ResNet50 and ResNet101. The VideoSwin model includes different backbone networks: Swin-T, Swin-S, Swin-B, and Swin-L. The MViTV2 model includes different backbone networks: MViTv2-S* and MViTv2-B*.

Table 5: Results on UCF-Sports1M. Average Source Only represents the average performance of the models from the source domain model zoo on the target domain. The best results are in bold.

| Method | U→ S | S→U | Avg |
|---|---|---|---|
| Source | 63.09 | 89.22 | 76.16 |
| SHOT | 71.35 | 96.18 | 83.77 |
| +LEEP* | 72.30 | 97.60 | 84.95 |
| +SUTE | 72.99 | 97.57 | 85.28 |
| STHC | 71.81 | 96.18 | 84.00 |
| +LEEP* | 71.80 | 97.60 | 84.70 |
| +SUTE | 76.16 | 97.57 | 86.87 |
| Decision | 71.36 | 96.44 | 83.90 |
| +LEEP* | 69.55 | 97.76 | 83.66 |
| +SUTE | 73.60 | 97.96 | 85.78 |
| Kd3A | 71.81 | 96.18 | 84.00 |
| +LEEP* | 67.9 | 96.40 | 82.15 |
| +SUTE | 76.27 | 97.57 | 86.92 |
| CAiDA | 71.81 | 96.31 | 84.06 |
| +LEEP* | 70.62 | 97.82 | 84.22 |
| +SUTE | 76.52 | 97.24 | 86.88 |
| Ours | **76.94** | **97.99** | **87.47** |

Table 6: Results on UCF-HMDB$_{full}$. Average Source Only represents the average performance of the models from the source domain model zoo on the target domain. The best results are in bold.

| Method | H→ U | U→H | Avg. |
|---|---|---|---|
| Source | 81.97 | 74.75 | 78.36 |
| SHOT | 93.45 | 83.14 | 88.30 |
| +LogME* | 96.78 | 81.83 | 89.31 |
| +SUTE | 96.80 | 86.67 | 91.74 |
| STHC | 93.98 | 82.87 | 88.43 |
| +LogME* | 96.27 | 81.20 | 88.74 |
| +SUTE | 95.56 | 86.08 | 90.82 |
| Decision | 93.30 | 83.14 | 88.23 |
| +LogME* | 97.83 | 76.23 | 87.03 |
| +SUTE | 96.16 | 87.40 | 91.78 |
| Kd3A | 93.24 | 82.83 | 88.04 |
| +LogME* | 96.20 | 81.16 | 88.68 |
| +SUTE | 95.54 | 85.98 | 90.76 |
| CAiDA | 92.90 | 82.38 | 87.64 |
| +LogME* | 96.31 | 78.68 | 87.50 |
| +SUTE | 95.52 | 84.91 | 90.22 |
| Ours | **99.57** | **91.29** | **95.43** |

Table 7: Results on Daily-DA. Source represents the average performance of the models from the source domain model zoo on the target domain. The best results are in bold.

| Method | M→ A | H→ A | A→ M | H→ M | A→ H | M→ H | Avg. |
|--------|------|------|------|------|------|------|------|
| Source | 42.49 | 35.07 | 27.50 | 39.08 | 36.34 | 57.92 | 39.73 |
| SHOT | 58.95 | 44.56 | 43.25 | 48.50 | 62.50 | 68.33 | 54.35 |
| +LEEP* | 63.98 | 46.10 | 47.50 | 35.75 | 59.58 | 65.83 | 53.12 |
| +LogME* | 41.91 | 32.36 | 27.00 | 47.75 | 34.17 | 61.67 | 40.81 |
| +MDE | 60.44 | 42.87 | 26.75 | 46.00 | 34.17 | 74.58 | 47.47 |
| STHC | 59.91 | 47.85 | 43.00 | 48.00 | 62.08 | 67.92 | 54.79 |
| +LEEP* | 65.33 | 51.08 | 47.50 | 37.75 | 65.00 | 67.08 | 55.62 |
| +LogME* | 46.39 | 36.60 | 17.50 | 47.25 | 35.83 | 62.50 | 41.01 |
| +MDE | 60.29 | 51.16 | 13.50 | 45.75 | 58.75 | 71.67 | 50.19 |
| Decision | 58.35 | 46.30 | 42.75 | 49.00 | 57.92 | 67.92 | 53.71 |
| +LEEP* | 63.88 | 45.66 | 47.50 | 29.00 | 57.92 | 65.42 | 51.56 |
| +LogME* | 38.94 | 32.53 | 27.25 | 48.50 | 35.42 | 62.50 | 40.86 |
| +MDE | 62.52 | 43.59 | 24.50 | 50.25 | 28.75 | 73.33 | 47.16 |
| Kd3A | 60.15 | 47.51 | 43.00 | 48.00 | 62.08 | 67.92 | 54.78 |
| +LEEP* | 64.9 | 51.60 | 45.25 | 37.50 | 59.17 | 67.08 | 54.25 |
| +LogME* | 46.53 | 37.01 | 26.75 | 47.25 | 35.00 | 62.08 | 42.44 |
| +MDE | 60.17 | 51.25 | 26.75 | 45.75 | 35.42 | 71.67 | 48.50 |
| CAiDA | 58.57 | 46.06 | 42.75 | 49.00 | 61.25 | 68.33 | 54.33 |
| +LEEP* | 66.14 | 49.91 | 46.50 | 28.00 | 58.75 | 67.08 | 52.73 |
| +LogME* | 40.37 | 41.23 | 26.00 | 48.75 | 32.08 | 63.33 | 41.96 |
| +MDE | 60.67 | 42.40 | 27.00 | 40.75 | 27.92 | 71.67 | 45.07 |
| Ours | **70.36** | **52.74** | **50.75** | 47.25 | **68.75** | **79.58** | **61.57** |

Table 8: Results on UCF-Sports1M. Average Source Only represents the average performance of the models from the source domain model zoo on the target domain. The best results are in bold.

| Method | U→ S | S→U | Avg |
|---|---|---|---|
| Source | 63.09 | 89.22 | 76.16 |
| SHOT | 71.35 | 96.18 | 83.77 |
| +LogME* | 51.6 | 93.12 | 72.36 |
| +MDE | 70.54 | 88.08 | 79.31 |
| +NMI | 67.86 | 96.4 | 82.13 |
| STHC | 71.81 | 96.18 | 84.00 |
| +LogME* | 53.27 | 93.02 | 73.17 |
| +MDE | 69.71 | 87.85 | 78.78 |
| +NMI | 67.73 | 96.4 | 82.07 |
| Decision | 71.36 | 96.44 | 83.90 |
| +LogME* | 45.63 | 93.87 | 69.75 |
| +MDE | 69.65 | 88.32 | 78.99 |
| +NMI | 69.19 | 96.27 | 82.73 |
| Kd3A | 71.81 | 96.18 | 84.00 |
| +LogME* | 53.25 | 93.02 | 73.14 |
| +MDE | 69.84 | 87.85 | 78.85 |
| +NMI | 67.9 | 96.4 | 82.15 |
| CAiDA | 71.81 | 96.31 | 84.06 |
| +LogME* | 61.54 | 90.57 | 76.06 |
| +MDE | 69.21 | 87.66 | 78.44 |
| +NMI | 68.17 | 96.62 | 82.40 |
| Ours | **76.94** | **97.99** | **87.47** |

Table 9: Results on UCF-HMDB$_{full}$. Average Source Only represents the average performance of the models from the source domain model zoo on the target domain. The best results are in bold.

| Method | H→ U | U→H | Avg. |
|---|---|---|---|
| Source | 81.97 | 74.75 | 78.36 |
| SHOT | 93.45 | 83.14 | 88.30 |
| +LEEP* | 86.69 | 83.56 | 85.13 |
| +NMI | 95.34 | 83.66 | 89.50 |
| +MDE | 85.77 | 86.16 | 85.97 |
| STHC | 93.98 | 82.87 | 88.43 |
| +LEEP* | 82.63 | 85.68 | 84.16 |
| +NMI | 92.95 | 81.45 | 87.20 |
| +MDE | 83.76 | 84.01 | 83.89 |
| Decision | 93.30 | 83.14 | 88.23 |
| +LEEP* | 85.40 | 84.32 | 84.87 |
| +NMI | 95.60 | 84.11 | 89.86 |
| +MDE | 85.38 | 86.29 | 85.84 |
| Kd3A | 93.24 | 82.83 | 88.04 |
| +LEEP* | 82.39 | 85.43 | 83.91 |
| +NMI | 92.89 | 81.29 | 87.09 |
| +MDE | 83.69 | 83.85 | 83.77 |
| CAiDA | 92.90 | 82.38 | 87.64 |
| +LEEP* | 94.11 | 86.02 | 90.07 |
| +NMI | 93.17 | 81.27 | 87.22 |
| +MDE | 89.01 | 84.19 | 86.60 |
| Ours | **99.57** | **91.29** | **95.43** |

