# OpenReview forum: "Multi-Model Induced Source-free Video Domain Adaptation"
_ICLR.cc/2025/Conference — ICLR 2025 Conference Withdrawn Submission_

### Official Review · Reviewer_hPy5 · 2024-11-03

**Soundness:** 3
**Presentation:** 2
**Contribution:** 2
**Rating:** 3
**Confidence:** 4

**Summary:**

This paper addresses the challenging problem of Source-Free Video Domain Adaptation (SFVDA), where the adaptation process relies solely on a pretrained model and unlabelled target videos, without access to labelled source data during adaptation. This constraint complicates effective feature alignment between domains. The authors highlight a common real-world scenario, which explores multiple models to conduct SFVDA.

A central challenge in MSFVDA is the effective aggregation of knowledge from multiple source models. To tackle this, the authors propose a novel approach called Multi-level Instance Transferability Calibration (MITC). MITC leverage existing transferability estimation metrics to evaluate the instance level.  To further enhance instance-level aggregation, the authors introduce an Instance-level Video Multi-Model Aggregation (IMVMA) method. IMVMA dynamically reweights the contribution of each model based on the distinctive characteristics of individual video instances, thus facilitating more precise and adaptable aggregation across diverse video contexts.

**Strengths:**

- The proposed method demonstrates strong empirical performance, achieving competitive results across several benchmark datasets, which attests to its effectiveness in handling source-free video domain adaptation.
- The motivation of leveraging multiple granularity to estimate overall transferability is clear

**Weaknesses:**

- Although the work targets the video domain adaptation problem, it lacks innovations specifically tailored to the unique characteristics of video data.
-The discussed limitation of existing transferability estimation methods lacks empirical results to support the discussion.
-The novelty of this work is somewhat limited, as many components, such as L_{SHTC}, transferability estimation metrics, SUTE, appear to be adaptations of methods from existing literature. This integration of existing techniques, while effective, leans more toward an engineering-focused contribution rather than offering substantial theoretical or methodological innovation.
- Given that L_{cor} is a primary contribution, a parameter sensitivity analysis for \theta_1 is conspicuously absent.
- There are several typos in this work. For example, Line 709 should be equation 9.

**Questions:**

Please see the weakness part.

---

### Official Review · Reviewer_Hn2C · 2024-11-03

**Soundness:** 3
**Presentation:** 2
**Contribution:** 2
**Rating:** 5
**Confidence:** 4

**Summary:**

The paper addresses a new task of Source-Free Video Domain Adaptation (SFVDA) by utilizing multiple pre-trained source models across different domains to improve the adaptability of video action recognition models for unlabeled target domains. It introduces a Multiple-Source-Video-Model Aggregation (MSVMA) framework, which uses two core modules: (1) Multi-level Instance Transferability Calibration (MITC) and (2) Instance-level Multi Video Model Aggregation (IMVMA). MITC estimates transferability at the instance level by integrating group- and dataset-level calibration to achieve accurate transferability across diverse models. IMVMA then leverages these instance-level transferability estimates to aggregate source models dynamically, resulting in an unsupervised method that achieves state-of-the-art performance on three benchmark datasets.

**Strengths:**

This work is original in extending SFVDA to multiple source models, creating a comprehensive aggregation framework. The authors address an underexplored area by enabling adaptation from multiple sources, which could improve adaptation quality significantly.

The introduction of MITC for multi-level instance transferability calibration is an effective way to tackle cross-model transferability, particularly with instance-level granularity that is uncommon in video domain adaptation.

The paper includes extensive experimentation across three diverse video datasets, demonstrating consistent improvements over several established baselines. Additionally, it uses various baselines that encompass both single-source and multi-source domain adaptation methods, enhancing the robustness of the findings.

The authors provide detailed ablation studies that carefully dissect the roles of each calibration level and weighting approach. These analyses highlight the significance of each component and validate their contributions quantitatively.

**Weaknesses:**

In some extent, the effectiveness of the proposed multi-model source-free domain adaptation method relies on the utilizing of multiple pretrained videos, which provide possibilities to improve the generalization ability of the source model. Therefore, one weakness of this paper is that it is unclear whether the effectiveness is from the proposed method or just from the using of multiple pretrained video classification models on the source domain.

The multi-model adaptation method SUTE [16] and multi-model aggregation at instance-level [17] are more related to this paper, since they also consider how to combine multiple models to address the domain shift. However, the result comparison in the experiment does not focus on these more similar methods, but focus on adjusting conventional source-free DA methods.

The writing should be improved. Some sentences are confusing. For example, in the abstract: “In this paper, we explore a new SFVDA setting where multiple source domains exist, each offering a library of source models with different architectures”. This paper only considers a single source domain with different pretrained models. Therefore, the sentence is misleading.

The reliance on hyperparameters, especially within the MITC calibration function and path generation network, may limit generalizability. The requirement for dataset-specific tuning to achieve optimal results reduces its utility in more generalized, adaptive scenarios.

The datasets used (Daily-DA, Sports-DA, UCF-HMDB) do cover various action recognition domains but may not sufficiently test the model’s adaptability to drastically different video types (e.g., low-light, surveillance, or atypical scenes). Testing the model’s resilience to these video types would help evaluate its versatility further.

The pathway generation network assigns instance-specific weights for model aggregation but lacks an interpretative mechanism to explain why certain models or instances are favored over others. Understanding these weight assignments could add insights into domain-specific transferability.

The heavy reliance on uncertainty metrics for transferability estimation may make the method sensitive to noise or outliers in the data, especially with diverse video content.

**Questions:**

How about the results of directly extending the existing image-based method SUTE [16] or [17] to the video task with the video feature extraction schemes used in this paper?

Could the authors elaborate on how the pathway generation network handles cases where multiple models have similar transferability scores? Does it introduce variability in performance if models are evenly weighted?

Are there plans to incorporate interpretability techniques to explain why certain instances or models receive higher transferability scores? Such explanations could aid in understanding the relationship between instance-specific characteristics and model selection.

The paper notes that aggregating too many models can reduce performance due to less transferable models. Would it be feasible to establish an optimal aggregation threshold across datasets, or does this threshold vary significantly by domain?

---

### Official Review · Reviewer_oQYN · 2024-11-03

**Soundness:** 1
**Presentation:** 2
**Contribution:** 2
**Rating:** 1
**Confidence:** 5

**Summary:**

This paper introduces the Instance-level Multi Video Model Aggregation (IMVMA) method to aggregate the different models for the task of source-free video domain adaptation with multiple models (model zoo), which is assisted by a MITC for instance-level transferability. The proposed IMVMA achieves good performance on several Unsupervised Video Domain Adaptation datasets.

**Strengths:**

Video (unsupervised) domain adaptation (UVDA/VUDA) has indeed gained more attention with the increase need for more generalizable and transferable video models for video analysis. Meanwhile, the task of source-free VUDA (SFVDA) with model zoo has indeed yet to be vastly explored despite the growing attention in SFVDA itself. Therefore, the explored task is of certain significance. The authors did also manage to provide comprehensive results on the various benchmarks which proves empirically that the proposed method are indeed empirically effective. Overall, the method can be followed even though there are some typos, and is empirically proven.

**Weaknesses:**

Despite the empirical success of the method, there are more concerns than strengths over the proposed methods. The authors should consider addressing the following issues:

Major:
1. The novelty of the method, specifically the novelty in the context concerning adaptation to the video domain, is very weak. While the concerned task is indeed considered novel for the video domain, it has, and as pointed out by the authors themselves, being addressed (at least to some extent) for the image domain. Therefore, the authors rely heavily on what has already been developed in the image domain and simply adopt it to the video domain. For instance, the MITC, though given a new name, is simply a combination of SUTE + entropy. Yet, despite the author did mention that adaptation in the video domain is "more challenging" "due to the inherent spatial and temporal complexities in the video", there is no evidence that the proposed method tackles the video domain differently. It is safe to say that the same method can simply be adopted to any domain (point cloud, time-series, NLP) with a simple change of model zoo. Rather than a MM-SFVDA task, it seems more like a method for MM-SFDA task, and due to the fact that there is little (or none) prior works on videos, the method is then adopted to the video domain. Since the authors tackle specifically for SF**V**DA not SFDA, the authors should emphasize how this method is different due to the characteristics of videos. Such strategy can be found in the various SFVDA papers such as [22] and [15] as cited in the paper.
2. While the authors did provide the supplementary code for revision, it is noted that the code is **NOT well anonymized**, and is not complete. While it is understandable that certain configuration files are missing as they can be filled by reading through the MMAction2 library (which is indeed taking a lot of time), it is not understandable and forgiveable that the authors simply leave out (whether by mistake or intentionally) personal information that hinders the double-blind process, which is to ensure that the review process is fair for all parties. This may not be technically related, but I highlight it in the Major part and will submit for further review.

Minor:
1. Some of the annotations are confusing and missing, for example, what is $\mathcal{H}$ in SUTE and the formulation for $\mathcal{T}_{I}$? What is $d$ in Equation 4 for group-level transferability?
2. There is no hyperparameter sensitivity analysis for $\theta_1$, which is the tradeoff parameter in the overall objective.
3. Why are the models (i.e., the 15 models including C3D, SlowFast, I3D and its variants, etc.) selected in the model zoo. Also it seems that the calibrated instance-weights perform much better in certain tasks such as MIT->ARID than the average weights , while sometimes it is comparable such as in H->A, why is there such an observable variation? What is the weight for the different models?
4. The literature review is not satisfactory and sometimes errorneous. E.g., in Line 64 when introducing SFVDA the author points to citation [14] which does not discuss SFVDA at all. SFVDA methods such as [a-c] are not included or discussed. Also, there have been a number of other methods for evaluating transferability (e.g., in [d-f]), why are these excluded?
5. There are some typos across the paper, please revise.

[a] Huang, Y., Yang, X., Zhang, J., & Xu, C. (2022, October). Relative alignment network for source-free multimodal video domain adaptation. In Proceedings of the 30th ACM International Conference on Multimedia (pp. 1652-1660).
[b] Zara, G., Conti, A., Roy, S., Lathuilière, S., Rota, P., & Ricci, E. (2023). The unreasonable effectiveness of large language-vision models for source-free video domain adaptation. In Proceedings of the IEEE/CVF International Conference on Computer Vision (pp. 10307-10317).
[c] Lo, S. Y., Oza, P., Chennupati, S., Galindo, A., & Patel, V. M. (2023). Spatio-temporal pixel-level contrastive learning-based source-free domain adaptation for video semantic segmentation. In Proceedings of the IEEE/CVF Conference on Computer Vision and Pattern Recognition (pp. 10534-10543).
[d] Yang, J., Qian, H., Xu, Y., Wang, K., & Xie, L. Can We Evaluate Domain Adaptation Models Without Target-Domain Labels?. In The Twelfth International Conference on Learning Representations.
[e] Tran, A. T., Nguyen, C. V., & Hassner, T. (2019). Transferability and hardness of supervised classification tasks. In Proceedings of the IEEE/CVF International Conference on Computer Vision (pp. 1395-1405).
[f] Ma, X., Gao, J., & Xu, C. (2021). Active universal domain adaptation. In Proceedings of the IEEE/CVF international conference on computer vision (pp. 8968-8977).

**Questions:**

As listed in the Weaknesses section.

**Details Of Ethics Concerns:**

The authors provide certain personal information (name) in the supplementary material, which is included in the code. See supplementary code under file code/IMVMA.py (which I believe is an important part of the code base), from lines 875 -- 895.

---

### Note · Authors · 2024-11-14

I have read and agree with the venue's withdrawal policy on behalf of myself and my co-authors.